# An Agent-Based Method for Feature Recognition and Path Optimization of Computer Numerical Control Machining Trajectories

**DOI:** 10.3390/s24175720

**Published:** 2024-09-03

**Authors:** Purui Li, Meng Chen, Chuanhao Ji, Zheng Zhou, Xusheng Lin, Dong Yu

**Affiliations:** 1Shenyang Institute of Computing Technology, Chinese Academy of Sciences, Shenyang 110168, China; lipurui22@mails.ucas.ac.cn (P.L.); chenmeng@sict.ac.cn (M.C.); jichuanhao22@mails.ucas.ac.cn (C.J.); zhouzheng18@mails.ucas.ac.cn (Z.Z.); zhimaoyi8@gmail.com (X.L.); 2University of Chinese Academy of Sciences, Beijing 100049, China; 3Shenyang CASNC Technology Co., Ltd., Shenyang 110168, China

**Keywords:** CNC system, intelligent elements, process analysis, path optimization, deep learning, feature recognition

## Abstract

In recent years, artificial intelligence technology has seen increasingly widespread application in the field of intelligent manufacturing, particularly with deep learning offering novel methods for recognizing geometric shapes with specific features. In traditional CNC machining, computer-aided manufacturing (CAM) typically generates G-code for specific machine tools based on existing models. However, the tool paths for most CNC machines consist of a series of collinear motion commands (G01), which often result in discontinuities in the curvature of adjacent tool paths, leading to machining defects. To address these issues, this paper proposes a method for CNC system machining trajectory feature recognition and path optimization based on intelligent agents. This method employs intelligent agents to construct models and analyze the key geometric information in the G-code generated during CNC machining, and it uses the MCRL deep learning model incorporating linear attention mechanisms and multiple neural networks for recognition and classification. Path optimization is then carried out using mean filtering, Bézier curve fitting, and an improved novel adaptive coati optimization algorithm (NACOA) according to the degree of unsmoothness of the path. The effectiveness of the proposed method is validated through the optimization of process files for gear models, pentagram bosses, and maple leaf models. The research results indicate that the CNC system machining trajectory feature recognition and path optimization method based on intelligent agents can significantly enhance the smoothness of CNC machining paths and reduce machining defects, offering substantial application value.

## 1. Introduction

The rapid advancement of artificial intelligence and machine learning technologies has established CNC technology as a crucial foundation in the field of intelligent manufacturing [1]. These emerging technologies have begun to directly influence the machining process, enhancing the accuracy of CNC systems, optimizing efficiency, bolstering competitiveness and sustainability, and significantly improving the quality of produced workpieces [2,3]. However, in current industrial practice, automated manufacturing heavily relies on data transfer between CNC machines and CAM systems. The machining programs generated directly by CAM systems typically focus only on the continuity of tool positions, often neglecting the finer details of the motion process. This can result in rough tool paths, insufficient smoothness, and sharp corners, thereby adversely affecting the machining process.

In the manufacturing process of most mechanical parts, various machining features are typically involved. Previously, these features had to be manually extracted and specified using traditional CNC feature recognition methods, which resulted in high labor intensity and were constrained by factors such as the operator’s skill level and the geometric shape of the workpiece. Consequently, this approach was limited to analyzing only simple or specific features. However, with the continuous development and upgrading of the manufacturing industry, traditional CNC systems have begun to face challenges in feature recognition. At this juncture, deep learning technology, with its ability to automatically learn and extract features, has significantly reduced the complexity and difficulty of the task [4]. Deep learning not only provides more precise feature recognition, benefiting from its robust data fitting and learning abilities, but it also adapts to more complex patterns and variations [5]. Its models can dynamically adjust parameters to accommodate new data or environments, a flexibility constrained in traditional CNC systems. Moreover, deep learning can handle large-scale data, gaining insights from vast amounts of information, which enables it to more accurately simulate and predict manufacturing processes, thereby facilitating intelligent decision-making and optimization.

Path optimization is paramount in modern manufacturing. It can substantially reduce waste and enhance efficiency and precision, thereby boosting competitiveness and sustainability. Effective path optimization minimizes ineffective reciprocation, ensures smooth and continuous machine tool movement, reduces wear and tear, extends equipment lifespan, and lowers energy consumption, thereby achieving green manufacturing [6,7,8]. Path optimization not only enhances machining efficiency but also significantly improves product quality. By reducing excessive machine movements and conflicts, unnecessary errors are avoided, machining precision is increased, and product quality is ensured. Furthermore, path optimization can flexibly accommodate various complex machining demands, enhancing the adaptability and flexibility of the manufacturing process and making it a critical component of modern intelligent manufacturing [9].

Despite significant progress in the aforementioned research, there remains room for improvement in feature recognition accuracy and the smoothness of path optimization. Therefore, to enhance production efficiency and product quality, adopting more efficient and intelligent methods is crucial. To address these challenges, we have developed a novel feature recognition method and path optimization algorithm. The primary contributions of this paper are summarized as follows:Introduction of an agent-based CNC system architecture, leveraging artificial intelligence technology to enhance the performance and efficiency of CNC systems.Proposal of a deep neural network integrating multiple neural network models and linear attention mechanisms for improved feature recognition efficiency.Utilization of four mechanisms to improve the COA in order to enhance the smoothness:
The honey badger algorithm initializes the coati population, thereby enhancing the initial population quality and optimization efficiency.Information such as the population size, iteration count, and fitness function is embedded into the improved path update rules, which allows the new rules to optimize the search process based on the current population status, thereby improving convergence speed.A dynamic multi-population strategy is used to comprehensively explore the search space and maintain population diversity.A gradient descent fitness-guided strategy dynamically adjusts the learning rate, thereby controlling the magnitude of each path point update for quicker convergence to the optimal solution.

In this study, we explore a novel method for CNC system process optimization, incorporating deep learning into CNC feature recognition and applying optimization algorithms to trajectory optimization. Our goal is to achieve automatic machining trajectory recognition and utilize artificial intelligence to make the trajectory optimization process more autonomous and intelligent, thereby enhancing the operational efficiency and precision of CNC systems to meet the rapid development needs of the manufacturing industry. This work will contribute to the advancement of CNC technology, providing new pathways for realizing more intelligent manufacturing.

The remainder of this paper is structured as follows. Section 2 provides a comprehensive review of related works on feature recognition and path optimization methods. Section 3 introduces the agent-based CNC system architecture. Section 4 presents the theory of machining feature recognition and the improved coati optimization algorithm. Section 5 showcases the experimental results of the deep learning network and path optimization. Finally, Section 6 concludes this paper.

## 2. Related Works

In the realm of numerical control, feature recognition and path optimization are two critical technical aspects. These technologies not only play a pivotal role in enhancing machining accuracy but also significantly improve the efficiency and automation level of the machining process. Through feature recognition technology, CNC systems can automatically identify the geometric features of workpieces, thus formulating more precise machining strategies. Concurrently, path optimization technology ensures that the tool moves along the optimal path during machining, minimizing processing time and energy consumption while enhancing surface quality and machining precision. The integration of these technologies renders the CNC machining process more intelligent and efficient, effectively reducing human error and improving product quality and production stability.

### 2.1. Feature Recognition in CNC Machining Based on Deep Learning

In the digital age, CNC (Computer Numerical Control) machining feature recognition has become increasingly important. Whether for machining parts or creating complex workpieces, this technology has significantly enhanced production efficiency and ensured product quality. It has had a profound impact on various fields, including computer-aided design (CAD), computer-aided process planning (CAPP), and computer-aided manufacturing (CAM). These three domains are crucial stages in modern engineering, design, and manufacturing, with feature recognition technology playing a vital supporting role in these tasks.

The CAD system enables users to perform intricate engineering design tasks using computer technology. The complexity of design primarily arises from the diversity in workpiece shapes and dimensions. Xu et al. [10] proposed an asynchronous LiDAR-camera fusion dynamic positioning framework based on deep clustering. By converting point clouds into distance images and introducing a neural network framework for drone recognition, they achieved robust positioning and identification of drones. However, this framework also has certain limitations. While converting point clouds into distance images improves processing efficiency, it may result in the loss of spatial information, thereby affecting the accuracy of three-dimensional model reconstruction. Deng et al. [11] introduced an innovative LiDAR-based framework for drone detection, positioning, and tracking. By combining the distance image projection, LiDAR point cloud depth clustering, and drone technology, they achieved a novel approach to drone detection. This framework also demonstrates unique application potential in modern intelligent manufacturing. Zhang et al. [12] proposed a deep learning network called BrepMFR, designed for feature recognition in B-rep models. This network effectively handles intersecting features of complex geometric structures. The advantage of this method lies in its ability to automatically detect intricate features, reducing manual intervention and enhancing design accuracy. However, a potential drawback of this approach is its high dependency on the quality of input data and computational complexity, which may limit its scalability in large-scale industrial applications.

Once the design is completed, the production process must be considered, in which CAPP plays a crucial role. CAPP encompasses a range of tasks, including determining material types, defining production steps, and setting machine parameters. In this process, feature recognition technology is vital, as it allows for appropriate settings and optimizations based on the specific features of the workpiece. Wang et al. [13] proposed DeepFeature, a hybrid learning framework based on Graph Neural Networks (GNNs) aimed at further enhancing the process planning stage. However, the complexity of GNN models may require substantial computational resources and specialized knowledge for effective implementation. Once all the features of the workpiece are accurately identified, precise machining programs can be generated, thereby optimizing the machining process, reducing costs, shortening production times, and improving product quality.

In the CAM domain, feature recognition technology can automatically transform design and process planning into tangible products, truly realizing intelligent manufacturing. Wu et al. [14] proposed a semi-supervised learning framework that leverages both labeled and unlabeled data to learn meaningful visual representations, providing a powerful tool for workpiece feature recognition. The primary advantage of this framework is its ability to utilize both labeled and unlabeled data, which is particularly valuable when labeled data are limited.

The aforementioned methods offer promising advancements in feature recognition, driving automation and efficiency improvements in manufacturing. Although deep learning provides greater accuracy and flexibility, its complexity and resource demands may restrict its practical application in certain scenarios. Nevertheless, ongoing progress in these fields is continually pushing the boundaries of intelligent manufacturing, making more cost-effective and high-quality production processes possible.

### 2.2. CNC Machining Path Optimization

In modern manufacturing, the significance of optimizing CNC machining paths is becoming increasingly prominent. Tool-path optimization techniques greatly enhance machining efficiency, reduce production costs, and improve product quality. The initial machining paths generated by CAD/CAM systems are typically formed into G-code based on design models and predetermined machining strategies. However, in practical applications, this process may introduce certain errors. Kaigom et al. [15] introduced a novel concept for developing RDT, which systematically captures robotic dynamics and purposefully gathers data. Through data analysis, it was discovered that the computational accuracy of CAD/CAM systems may be limited by current hardware and software conditions, leading to potential issues with the smoothness of generated tool paths. These problems may cause the machine to experience sudden stops and restarts during operation, thereby increasing wear and reducing its lifespan. Discontinuous tool paths may also adversely affect machining accuracy, resulting in inconsistent product quality.

Fang et al. [16] proposed a real-time smoothing algorithm called g-smooth, which ensures higher-order continuity of tool paths in three-axis hybrid machining. This algorithm addresses the discontinuities in the initial tool paths, enhancing the stability of machine operations. The primary advantage of Fang’s method is its ability to maintain tool-path continuity, thereby reducing machine vibration and wear. However, the real-time nature of this algorithm may impose a computational burden, potentially limiting its applicability in highly complex machining scenarios. To address the continuity issues of mixed G01 and G02/03 encoding, Shi et al. [17] proposed a comprehensive tool-path smoothing method based on a motion overlap strategy, employing a heuristic algorithm to seek optimal motion parameters that meet the constraints of mixed encoding and efficiency requirements. However, when dealing with extremely complex geometries, the heuristic algorithm may converge to suboptimal solutions, thereby affecting overall machining accuracy. For particularly intricate 3D models, CAD/CAM systems may struggle to generate an optimal initial machining path. In such cases, the resulting G-code may exhibit discontinuities or excessive spacing at corners, leading to severe machine vibration during sharp turns, which negatively impacts machining accuracy and efficiency. Hua et al. [18] proposed an effective solution by adjusting the discrete curvature at each tool-tip point to refine the tool-tip position, thereby reducing the maximum curvature of the path and effectively mitigating vibration at sharp corners. The main advantage of Hua’s method lies in its focus on curvature adjustment, particularly in reducing tool vibration at critical points. Zhang et al. [19] introduced an adaptive tool-path generation method that employs a dual-head snake algorithm based on the least squares method to smooth the original part contours, generating tool paths composed of straight lines and arcs. This method excels in reducing machine vibration, improving accuracy, and enhancing efficiency. However, its reliance on the least squares method may introduce limitations in capturing highly complex surface features, potentially affecting the smoothness of the tool path in such cases.

The aforementioned studies collectively address the issue of tool-path discontinuity, making the movement paths of tools during machining more fluid. By reducing vibrations caused by sharp turns, these methods contribute to improved machining accuracy and efficiency. However, when selecting the most suitable tool-path optimization strategy, the trade-off between computational complexity and real-time applicability remains a crucial consideration.

## 3. Agent-Based CNC System Architecture

In recent years, with the continuous development of the manufacturing industry, CNC machining technology has played a crucial role in enhancing production efficiency and machining precision. However, traditional CNC path-planning methods often rely on preset rules and algorithms, which struggle to adapt to complex and dynamic machining environments. To address this issue, an agent-based CNC system architecture is proposed, encompassing the modeling and driving of intelligent CNC. This includes the digital twin of intelligent CNC and machine tool machining and considers the application of deep learning technology in feature recognition and path optimization within CNC machining, providing practical references for achieving intelligent CNC.

### 3.1. Intelligent Requirements

Intelligent manufacturing systems achieve optimal manufacturing outcomes by integrating the capabilities of machines, humans, and processes, thereby optimizing the use of manufacturing resources, adding value to people’s lives and careers, and reducing waste. Thus, intelligent manufacturing systems hold significant importance in modern societal development [20]. CNC systems, as the core component of CNC technology, play a vital role in CNC machining. Intelligent CNC systems offer higher programmability and flexibility, significantly improving manufacturing quality and efficiency [21]. CNC machines control the movement and machining processes of machine tools through CNC systems, executing automated machining based on pre-written program instructions.

The servo system is a critical part of CNC machines, comprising servo motors, drivers, and feedback devices. It is crucial for coordinating the trajectory accuracy between multiple feed axes and ensuring contour precision [22]. Digital twin technology is a potential solution for enhancing automation and advancing toward intelligent manufacturing [23]. By utilizing data collected from sensors, digital twin technology can monitor the operational status of CNC machines and servo systems in real time and generate their real-time digital models. Feature recognition is a key issue in intelligent manufacturing, enabling the extraction of valuable geometric information from solid models and achieving seamless digital connectivity from design to manufacturing [13]. For identified defective paths, optimization algorithms are employed to enhance path quality.

In traditional manufacturing modes, design and manufacturing are separate, with tool-path defects only fed back after manufacturing flaws are detected, requiring extensive effort to adjust and redesign the model. This paper proposes a new workflow, as shown in Figure 1. The red dashed box indicates the newly added workflow. After the CAM system generates the initial tool trajectory, feature recognition and optimization are performed to generate high-quality, smooth paths for subsequent manufacturing.

### 3.2. System Model and Structure

A modular structure plays a critical role in the design of CNC systems. The synergy between various intelligent components across modules fosters a dynamic environment for learning and self-optimization, as illustrated in Figure 2. The intelligent elements include the following:CAD/CAM: The process of design and manufacturing using NX software (Version 12.0, Siemens Digital Industries Software, Plano, TX, USA). This is the input part of the intelligent system, providing design data and manufacturing instructions.Learning: Extracts useful information from data to optimize models and manufacturing processes.Digital Twin: Provides a virtual environment for testing and optimization, enhancing efficiency and precision.Sense: Monitors various parameters during the manufacturing process, providing real-time feedback to the digital twin and optimization modules.Optimization: The process of optimizing system performance based on learning and sensing data, reducing resource consumption, and refining manufacturing processes.NC System and Machine Tool: Receives instructions from the optimization module and performs machining and manufacturing according to CNC system directives.

The intelligent CNC model encompasses the entire process from model design to final mechanical processing, including real-time feedback. This model begins with NX software, which is crucial in both the design and manufacturing stages. Achieving the required precision and surface quality is mainly addressed by selecting the tool-path density and appropriate cutting conditions. However, these measures are not always sufficient, and even with optimal choices, the surface quality and precision often decline [17]. By utilizing deep learning technology for feature extraction and pattern recognition, a seamless digital connection from design to manufacturing can be established based on the concept of features [13]. The optimization stage is the critical link in the entire process, focusing on optimizing the process flow based on data analyzed by the learning module. For identified rough paths, further calculations and adjustments through optimization algorithms are necessary to enhance path efficiency and precision. To achieve smooth and continuous machine motion, local corners must be smoothed to ensure the continuity and smoothness of the machining path [6]. Digital twins represent virtual replicas of the physical manufacturing environment, providing new opportunities for real-time monitoring of the machining process. They can simulate the actual production environment while considering changes in the machining process and operating conditions [24]. This enables testing and validation in a virtual environment, thereby reducing risks in actual production.

The sensing module involves sensors and other devices that collect real-time data from physical machines, monitoring various parameters in the production process and feeding data back to the system. The CNC system, as the brain of the machine tool [25], needs to control the machine based on inputs from the optimization and sensing modules to achieve precise control of the machine tool. The machine tool, being the physical apparatus that performs manufacturing tasks, plays a critical role in manufacturing, as its performance significantly impacts product quality and production efficiency [26]. Hence, intelligent modules need to be added to facilitate communication. The information flow process begins with CAD/CAM design, inputting the design into the learning module, which processes the data and inputs it into the optimization module. The optimization module communicates with the digital twin for simulation and validation processes, and the digital twin returns data to the learning module for further optimization. The sensing module collects real-time data from physical machines and sends it to the digital twin and learning module. The optimized and validated process is sent to the NC system, which controls the machine tool to execute manufacturing tasks, and the data from the machine tool are collected and fed back by the sensing module, completing the feedback loop.

The proposed system model demonstrates the interaction and data flow among various components in an intelligent manufacturing system, emphasizing a continuous loop of learning, optimization, and real-time feedback to improve manufacturing processes. Through this closed-loop system, the manufacturing process becomes more intelligent, automated, and efficient, thereby enhancing production quality and efficiency while reducing costs and risks.

### 3.3. Assembly Line Work Mode

In this study, the design of the intelligent agent module revolves around the CNC system processing workflow, as illustrated in Figure 3. The intelligent agent module acts as a critical node in the integrated processing chain of the CNC system, facilitating information transmission and processing, thereby achieving more efficient and precise perception and control of the machining process. Manufacturing activities start with the CAD/CAM design stage, including modeling and CNC machining. Next is the post-processing stage, involving tool-path generation and G-code generation. Subsequently, the CNC system receives these instructions and performs instruction interpretation and data processing. At this stage, process and data analysis are also conducted.

To evaluate the machining process, this study employs feature recognition technology based on deep learning neural networks to analyze and assess the results. The optimization stage utilizes optimization algorithms and visualization tools to improve the machining process, ensuring operational efficiency and precision. The process then moves into the servo control and machine tool machining stage, where actual machining operations and axis movements occur, along with motion control and signal conversion. Digital twin technology is used for feedback and improvement of the entire process, dynamically adapting to the actual machining requirements and conditions by combining virtual and real elements, thus achieving comprehensive monitoring and optimization of the machining process. This information flow is not unidirectional but aids in the dynamic adaptation of the system, better aligning with actual machining requirements and conditions.

## 4. Mathematical Model

In this section, the feature design of machining trajectories and the fundamental procedure of the improved COA are introduced.

### 4.1. Machining Path Feature Design

By analyzing the physical characteristics of the control system’s machining paths, feature extraction and recognition are performed for paths exhibiting uneven smoothness. The features of the machining paths are defined below.

A path consists of a series of coordinates $\left(x_{i},y_{i}\right)$. Given that the contribution of the *z*-axis coordinates is minimal, the path is identified using its two-dimensional coordinates. For each point *i* on the path, the direction vector $v_{i}$ is defined as the vector from point *i* to point i+1: $v_{i}=\left(x_{i+1}-x_{i},y_{i+1}-y_{i}\right)$.

The dot product of two vectors, $v_{i}$ and $v_{i-1}$, is defined by Equation (Equation 1):(1)$$v_{i}\cdot v_{i-1}=\left(x_{i+1}-x_{i}\right)\left(x_{i}-x_{i-1}\right)+\left(y_{i+1}-y_{i}\right)\left(y_{i}-y_{i-1}\right)$$

The formula defines the dot product of two vectors, $v_{i}$ and $v_{i-1}$, to quantify the directional relationship between adjacent vectors.

The Euclidean length of vector $v_{i}$ is defined by Equation (Equation 2):(2)$$|v_{i}|=\sqrt{{\left(x_{i+1}-x_{i}\right)}^{2}+{\left(y_{i+1}-y_{i}\right)}^{2}}$$

This formula computes the Euclidean distance between adjacent points, ensuring that the measurement of vectors aligns with the geometric shape of the actual path.

To quantitatively represent the local curvature of a path, denoted as LocalCurvature, a cumulative value of angle changes between all adjacent vectors from the first point to the *n*-th point, is defined. The formula for the local curvature is given in Equation (Equation 3):(3)$$\mathrm{LocalCurvature}=\sum_{i=1}^{n-2}\left(1-\frac{v_{i}\cdot v_{i-1}}{|v_{i}\left|\right|v_{i-1}|}\right)$$
where:$v_{i}\cdot v_{i-1}$ is the dot product of vectors $v_{i}$ and $v_{i-1}$.$|v_{i}|$ and $|v_{i-1}|$ are the Euclidean lengths of vectors $v_{i}$ and $v_{i-1}$ respectively.

The formula reflects the curvature variation of the path by calculating the angle between adjacent vectors. The dot product $v_{i}\cdot v_{i-1}$ is normalized by the product of the Euclidean lengths of the vectors, yielding the cosine value between the two vectors, within the range [−1,1]. A cosine value closer to 1 indicates that the two vectors are more aligned in the same direction, suggesting lower curvature. By summing $1-\frac{v_{i}\cdot v_{i-1}}{|v_{i}\left|\right|v_{i-1}|}$, the overall local curvature of the path is obtained. Thus, a higher LocalCurvature value signifies a less smooth path composed of *n* points, whereas a lower value indicates a smoother path.

In the case of a set number of path points (n=50), the local curvature of different paths was categorized, as shown in Table 1. This experimental setup ensures a sufficient density of path points, allowing the variations in local curvature to effectively reflect the oscillatory characteristics of the path.

After feature classification is completed, the feature points and their labels are used to train the MCRL network model for the automatic recognition of features in new, unlabeled samples. This model learns to recognize the smoothness of paths from the input coordinates. The workflow for the described machining path feature recognition is illustrated in Algorithm 1.
**Algorithm 1** Path Categorization Algorithm Based on LocalCurvature**Require:** *A*: A set of points $\left\{a_{1},a_{2},\ldots ,a_{n}\right\}$, where $a_{i}=\left(x_{i},y_{i}\right)$;**Ensure:** LabelPath: A path with a feature label;  1:The point set *A* is grouped into groups of 50 points and stored in the Group list: $\left[g_{1},g_{2},\ldots ,g_{n}\right]$, where $g_{i}=\left[p_{1},p_{2},\ldots ,p_{50}\right]$, $p_{i}=\left(x_{i},y_{i}\right)$;  2:**for** each group *g* in Group **do**  3:   **for** each point *p* in *g* **do**  4:     Generate initial set of vectors $V_{i}$: $\left\{v_{1},v_{2},\ldots ,v_{49}\right\}$, where $v_{i}=\left(x_{i+1}-x_{i},y_{i+1}-y_{i}\right)$;  5:     Compute the local curvature of $V_{i}$: $LocalCurvature=\sum_{i=2}^{49}\left(1-\frac{v_{i}\cdot v_{i-1}}{|v_{i}\left|\right|v_{i-1}|}\right)$;  6:     **if** LocalCurvature≤1.9 **then**  7:        Label $V_{i}$ as “Category 1”;  8:     **else if** 1.9<LocalCurvature<3.5 **then**  9:        Label $V_{i}$ as “Category 2”;10:     **else if** 3.5<LocalCurvature<7.6 **then**11:        Label $V_{i}$ as “Category 3”;12:     **else**13:        Label $V_{i}$ as “Category 4”;14:     **end if**15:   **end for**16:**end for**17:LabelPath = {all labeled paths from steps 17 and 18};18:LabelPath was used to train the MCRL model;19:**return** LabelPath

The algorithm aims to classify segments of the path based on local curvature and provide labeled data for subsequent model training. Initially, the algorithm divides the input point set *A* into several groups, each containing 50 consecutive points, with each point represented by its coordinates $\left(x_{i},y_{i}\right)$. These groups are stored in a list Group.

For each group *g* in the list Group, the algorithm processes each point *p* sequentially. During the processing of each point, the algorithm generates an initial vector set $V_{i}=\left\{v_{1},v_{2},\cdots ,v_{49}\right\}$ consisting of 49 vectors, where each vector $v_{i}$ represents the coordinate difference between adjacent points. Subsequently, the algorithm calculates the local curvature LocalCurvature of the vector set $V_{i}$. The formula for local curvature is given by
$$\mathrm{LocalCurvature}=\sum_{i=2}^{49}\left(1-\frac{v_{i}\cdot v_{i-1}}{|v_{i}\left|\right|v_{i-1}|}\right)$$

Based on the computed local curvature, the algorithm classifies the path segments into different categories. If LocalCurvature is less than or equal to 1.9, the segment is labeled as “Category 1”; if LocalCurvature is between 1.9 and 3.5, it is labeled as “Category 2”; if LocalCurvature is between 3.5 and 7.6, it is labeled as “Category 3”; and if LocalCurvature is greater than or equal to 7.6, it is labeled as “Category 4”.

Finally, all labeled path segments are aggregated into a labeled path LabelPath. This labeled path is then used to train the MCRL model. By precisely calculating local curvature, the algorithm effectively classifies the path segments, providing clear input data for subsequent machine learning model training.

### 4.2. Path Optimization Design

The coati optimization algorithm (COA) [27] is a biomimetic algorithm inspired by the foraging behavior of South American coatis. During their foraging, coatis mark their paths using visual and olfactory cues and share location information. These markers fade over time, but their concentration remains higher on shorter paths. Thus, the coati group can detect these markers and select an optimal path, moving toward areas with stronger signals.

This section proposes four mechanisms to overcome the shortcomings of the traditional coati algorithm. Firstly, the honey badger algorithm (HBA) is introduced to initialize the coati population, enhancing optimization efficiency by increasing population diversity. Secondly, an improved heuristic function is proposed to enhance the algorithm’s goal orientation, effectively reducing randomness in the search. Thirdly, a dynamic multi-population strategy is introduced to prevent the algorithm from falling into local optima, thereby enhancing global search capabilities. Finally, a gradient descent fitness-guided strategy is proposed to accelerate convergence. These four mechanisms are combined to form a new algorithm, referred to as the novel adaptive coati optimization algorithm (NACOA).

#### 4.2.1. Honey Badger Algorithm for Population Initialization

The initialization of the coati population may have several shortcomings: despite optimizing the selection process, the quality of the initial population may be influenced by the initial input data and parameter settings, leading to unstable results. If the individuals in the initial dataset exhibit limited genetic diversity, the algorithm may struggle to overcome this limitation, ultimately resulting in insufficient genetic diversity within the population. The algorithm’s effectiveness depends on high-quality, comprehensive initial data. If the data are inaccurate or incomplete, it may affect the algorithm’s performance and the effectiveness of population initialization.

To address these issues, the honey badger algorithm (HBA) [28] is used to initialize the coati population. This algorithm mimics the searching and hunting behavior of honey badgers, optimizing individual selection and distribution strategies to ensure that the population consists of highly adaptable and survivable individuals. The HBA has dynamic adjustment capabilities, allowing it to continuously optimize initialization strategies based on the actual population situation and adapt to changes in different environments and conditions.

Moreover, the HBA has high computational efficiency, enabling it to find optimal solutions in a relatively short time, thereby reducing human intervention and decision-making time. By automating selection and optimization processes, the algorithm reduces human bias and errors, enhancing the scientific rigor and rationality of population initialization. The HBA can also simulate individual adaptability under different environmental conditions, helping to select individuals best suited to new environments and thus improving the overall adaptability of the population.

The basic process of generating the coati initialization population using the honey badger algorithm is as follows:
**a** **Initialization phase**

Generate an initial honey badger population and optimize it to obtain a more optimal initial population. The steps are as follows:*i* *Generate initial honey badger population:*A two-dimensional point set path = $\left\{P_{1},P_{2},\ldots ,P_{n}\right\}$ represents a set of points on the path, where $P_{i}=\left(x_{i},y_{i}\right)$.Set the start and end points: $start\_point=P_{1}$ and $end\_point=P_{n}$.Randomly insert points: Randomly select the remaining points and randomly insert them into a certain position on the path until all points are inserted.

*ii* 
*Initialize each individual $p_{k}$ in the population:*


$p_{k}=\left\{P_{1},P_{k,2},P_{k,3},\ldots ,P_{k,n-1},P_{n}\right\}$, where $\left\{P_{k,2},P_{k,3},\ldots ,P_{k,n-1}\right\}$ are random permutations of $\left\{P_{2},P_{3},\ldots ,P_{n-1}\right\}$, and the population size is pop_size=N.

**b** 
**Defining intensity**


Define the intensity $I_{i}$, as shown in Equation (Equation 4):(4)$$\left\{\begin{array}{c} I_{i}=r_{1}\times \frac{S}{4\pi d_{i}^{2}}\\ S=\left|p_{k}\left[i+1\right]-p_{k}\left[i\right]\right|\\ d_{i}=\left|g_{best}\left[i\right]-p_{k}\left[i\right]\right| \end{array}\right.$$
where:$r_{1}$ is a random number, uniformly distributed in (0,1).*S* is the intensity.$p_{k}\left[i\right]$ is the *i*-th point of individual *k*.$g_{best}\left[i\right]$ is the *i*-th point of the global best position.$d_{i}$ represents the distance between the global best position and the current point.

The odor intensity $I_{i}$ is a function of the distance $d_{i}$ between the current position and the global best position $g_{\mathrm{best}}\left[i\right]$. This model is inspired by the olfactory foraging mechanism, where the intensity decreases with distance, guiding agents to move toward more promising areas.

**c** 
**Simulation of honey badger foraging behavior**


The process of updating positions is divided into two parts—the “digging phase” and the “foraging phase”—as shown in Equation (Equation 5):(5)$$p_{k}{\left[i\right]}^{\prime }=\left\{\begin{array}{cc} g_{\mathrm{best}}\left[i\right]+\beta \times I\times g_{\mathrm{best}}\left[i\right]+r_{2}\times \alpha_{1}\times d_{i}\times \left|\cos\left(2\pi r_{3}\right)\times \left[1-\cos\left(2\pi r_{4}\right)\right]\right| & \mathrm{if}\;r_{6}\leq 0.5\\ g_{\mathrm{best}}\left[i\right]+r_{5}\times \alpha_{1}\times d_{i} & \mathrm{else} \end{array}\right.$$
where:$\alpha_{1}$ is a constant.β is a constant indicating the honey badger’s ability to obtain food.$p_{k}{\left[i\right]}^{\prime }$ represents the updated position.$r_{2},r_{3},r_{4},r_{5},r_{6}$ are random numbers between 0 and 1.

This dual-phase strategy enhances the algorithm’s ability to refine the search in promising regions while maintaining a global perspective to avoid local optima.

**d** 
**Fitness function**


*i* 
*Curvature calculation*


Calculate the curvature of every three adjacent points, as shown in Equation (Equation 6):(6)$$curvature_{i}=1-\frac{v_{i}\cdot v_{i-1}}{\left|v_{i}\right|\left|v_{i-1}\right|}$$
where $v_{i}=\left(x_{i+1}-x_{i},y_{i+1}-y_{i}\right)$.

The total curvature is given by Equation (Equation 7):(7)$$LocalCurvature=\sum_{i=2}^{n-1}curvature_{i}$$

*ii* 
*Smoothness penalty*


The smoothness penalty is given by Equation (Equation 8):(8)$$SmoothnessPenalty=\sum_{i=2}^{n-1}second\_diff_{i}$$
where the second difference $second\_diff_{i}$ is calculated using Equation (Equation 9):(9)$$second\_diff_{i}=\left|\right|P_{i+1}-2P_{i}+P_{i-1}\left|\right|=\sqrt{{\left(x_{i+1}-2x_{i}+x_{i-1}\right)}^{2}+{\left(y_{i+1}-2y_{i}+y_{i-1}\right)}^{2}}$$

*iii* 
*Fitness function*


The fitness function is given by Equation (Equation 10):(10)$$Fitness(p_{k})=LocalCurvature+SmoothnessPenalty$$

The adaptive function comprises the local curvature and smoothness penalty terms. The curvature term ensures that the generated path is smooth and feasible, while the smoothness penalty refines the path by minimizing abrupt changes. This combination optimizes the overall quality of the path, making the algorithm suitable for complex optimization problems.

**e** 
**Updating the honey badger population**


Iteratively update each individual in the population, searching for better paths. In each iteration, update the paths and calculate their fitness to find the current best path. The update rule is given by Equation (Equation 11):(11)$$g_{best}=\arg\min_{p_{k}\in population}Fitness\left(p_{k}\right)\;\mathrm{over}\;\mathrm{generations}$$

This ensures that the algorithm continuously improves the solution quality while maintaining the diversity necessary to avoid premature convergence.

**f** 
**Initialization population of COA**


Use the population generated by the HBA as the initial population for the COA.

Through the above steps, the HBA can effectively enhance the quality and efficiency of the initial population.

#### 4.2.2. Enhanced Path Update Rule

In the coati optimization algorithm, the fixed parameter $\alpha_{2}$ cannot adapt dynamically during the search process, resulting in the inability to balance between global and local searches to find the optimal solution. This might lead to a decline in search efficiency. Therefore, we adopt $\alpha_{2}$, $r_{7}$, and $r_{8}$ to balance the global and local search, as illustrated in Equation (Equation 12):(12)$$p_{k}{\left[i\right]}^{\prime }=p_{k}\left[i\right]+\alpha_{2}\cdot r_{7}\cdot \left(g_{\mathrm{best}}\left[i\right]-p_{k}\left[i\right]\right)+\alpha_{2}\cdot r_{8}\cdot \left(\mathrm{rand}-0.5\right)$$
$\alpha_{2}$ is a factor that dynamically adjusts with the algebra *g*, as shown in Equation (Equation 13):(13)$$\alpha_{2}=\alpha_{\max}-\left(\alpha_{\max}-\alpha_{\min}\right)\cdot \frac{g}{G}$$
where:$\alpha_{\max}$ is the initial maximum value of $\alpha_{2}$.$\alpha_{\min}$ is the minimum value of $\alpha_{2}$.*g* is the current iteration number, incrementing from 1 to *G*.*G* is the total number of iterations in the algorithm.

$r_{7},r_{8}$, population fitness standard deviation std_dev, and average population fitness $\bar{F}$ are defined as shown in Equation (Equation 14):(14)$$\left\{\begin{array}{c} r_{7}=\mathrm{rand}\cdot \left(1+\frac{\mathrm{std}\_\mathrm{dev}}{\bar{F}}\right)\\ r_{8}=\mathrm{rand}\cdot \left(1+\frac{\mathrm{std}\_\mathrm{dev}}{\bar{F}}\right)\\ \mathrm{std}\_\mathrm{dev}=\sqrt{\frac{1}{N}\sum_{k=1}^{N}{\left(\mathrm{Fitness}\left(p_{k}\right)-\bar{F}\right)}^{2}}\\ \bar{F}=\frac{1}{N}\sum_{k=1}^{N}\mathrm{Fitness}\left(p_{k}\right) \end{array}\right.$$

In the early stages of the algorithm, a higher adaptive $\alpha_{2}$ encourages extensive exploration, covering a broader search space and aiding in escaping local optima. As the algorithm progresses, $\alpha_{2}$ gradually decreases, promoting fine-tuned exploitation and assisting in converging to the optimal solution. The adaptive $\alpha_{2}$ enhances the flexibility and precision of the search, leading to higher solution quality, closer approximation to the global optimum, and faster convergence.

By employing standard deviation-based adjustments for $r_{7}$ and $r_{8}$, the current population’s state can be leveraged to optimize the search process. In regions with higher fitness, the search efficiency increases, accelerating convergence and reducing the iterations needed to reach the optimal solution. In areas of lower diversity, random perturbations help avoid local optima, improving the likelihood of discovering the global optimum.

By employing the above methods, the algorithm effectively balances global search and local refinement, enhancing the likelihood of finding the global optimum while reducing computation time.

#### 4.2.3. Dynamic Multi-Population Strategy

In the coati optimization algorithm, as individuals converge prematurely near local optima, the population may lack diversity, making it difficult to find the global optimum. A single population strategy can lose the balance between exploration (searching for new solutions) and exploitation (using known good solutions), resulting in a limited search space. Hence, employing a dynamic multi-population strategy can enhance the optimization process by dividing the population into multiple sub-populations. Each sub-population searches in different areas, increasing the probability of finding the global optimum. The steps are as follows:**a** **Multi-Population Initialization**

Divide the population into $M_{0}$ sub-populations, each containing $N_{g}$ individuals:$$\mathrm{population}=\{{\mathrm{sub}\_\mathrm{pop}}_{1},{\mathrm{sub}\_\mathrm{pop}}_{2},\ldots ,{\mathrm{sub}\_\mathrm{pop}}_{M_{0}}\}$$
where ${\mathrm{sub}\_\mathrm{pop}}_{a}$ represents the *a*-th sub-population, the total population size is *N*, and the size of each sub-population ${\mathrm{sub}\_\mathrm{pop}}_{a}$ is given by Equation (Equation 15):(15)$$N_{g}=\frac{N}{M_{0}}$$

**b** 
**Updating Sub-Population Size**


At generation *g*, the number of current sub-populations $M_{g}$ is dynamically adjusted to accommodate the evolutionary characteristics of the population and the convergence requirements of the algorithm, thereby improving efficiency and solution quality. The adjustment rule is given by Equation (Equation 16):(16)$$M_{g}=\max\left(M_{\min},\min\left(M_{\max},M_{0}-\left\lfloor \frac{g\cdot (M_{0}-M_{\min})}{G}\right\rfloor \right)\right)$$
where:$M_{\min}$: Minimum sub-population size$M_{\max}$: Maximum sub-population size

**c** 
**Selection of the Optimal Individual**


Within each sub-population ${\mathrm{sub}\_\mathrm{pop}}_{a}$, identify the individual with the minimum fitness value, as expressed in Equation (Equation 17):(17)$$p_{{\mathrm{best}}_{a}}=\min_{p_{k}\in {\mathrm{sub}\_\mathrm{pop}}_{a}}\mathrm{Fitness}\left(p_{k}\right)$$
where $k\in \{1,\ldots ,\frac{N}{M_{0}}\}$.

Within each sub-population, we select the best-performing individual $P_{\mathrm{best}}$ by comparing individual fitness values. This is crucial for guiding the evolutionary direction of the sub-population, ensuring that the algorithm continuously evolves toward solutions with improved optimization performance.

The global optimal individual is denoted by Equation (Equation 18):(18)$$g_{\mathrm{best}}=\min_{a\in \{1,\ldots ,M_{0}\}}p_{{\mathrm{best}}_{a}}$$

**d** 
**Individual Exchange Strategy**


Every *T* iterations, perform an exchange of individuals between sub-populations. Let $p_{k}$ and $p_{t}$ be individuals randomly chosen from sub-populations ${\mathrm{sub}\_\mathrm{pop}}_{a}$ and ${\mathrm{sub}\_\mathrm{pop}}_{b}$, respectively. The exchange rule is illustrated in Equation (Equation 19):(19)$$\left\{\begin{array}{c} {\mathrm{sub}\_\mathrm{pop}}_{a}={\mathrm{sub}\_\mathrm{pop}}_{a}\cup \left\{p_{t}\right\}\setminus \left\{p_{k}\right\}\\ {\mathrm{sub}\_\mathrm{pop}}_{b}={\mathrm{sub}\_\mathrm{pop}}_{b}\cup \left\{p_{k}\right\}\setminus \left\{p_{t}\right\} \end{array}\right.$$

In the early stages of the algorithm, the number of sub-populations is approximately $M_{0}=10$, each with a size of $\frac{N}{M_{0}}=100$ individuals. This larger number of sub-populations maintains diversity and helps to explore a broader search space, thus preventing premature convergence to local optima. During the middle stages, the number of sub-populations gradually decreases to approximately $M_{g}=6$, each with a size of $\frac{N}{6}\approx 167$ individuals. As the number of sub-populations decreases, the search transitions to a more focused phase, balancing exploration and exploitation. Adjusting the sub-population size dynamically based on iterations ensures that the algorithm can efficiently search in appropriate regions, enhancing flexibility and reducing the risk of local optima entrapment through inter-sub-population exchanges every T=10 iterations. In the final stages, the number of sub-populations decreases to $M_{\min}=2$, each with a size of $\frac{N}{2}=500$, concentrating on fine-tuning the optimal solutions and thereby improving the final solution quality.

This strategy ensures the discovery and utilization of global optima by effectively escaping local optima in a vast search space. By dynamically adjusting the sizes of sub-populations and the exchange strategies, the algorithm not only enhances adaptability to complex problems but also optimizes computational resource efficiency. Theoretically, this method significantly reduces computation time and improves solution quality. The dynamic adjustment of sub-population size, based on the population’s performance, iteratively optimizes the balance between exploration and exploitation, thereby overcoming local optima constraints in the later stages of the algorithm. The individual exchange strategy strengthens the flow of information and genetic diversity within the population, aiding in the discovery of potential optimal regions that are not yet covered by the current population.

#### 4.2.4. Gradient Descent-Based Adaptive Guidance Strategy

In the coati optimization algorithm, the optimization process relies heavily on global search and random perturbations. This broad search in the solution space may slow down the convergence rate due to a lack of precise local tuning, ultimately leading to suboptimal solutions in complex solution spaces. To address this issue, a gradient descent-based adaptive guidance strategy can be employed. This approach utilizes gradient information for fine-tuning local adjustments, swiftly steering the search toward the optimal solution. The strategy provides directional information, guiding how to adjust points on the path, reducing both total curvature and gradient penalties. This helps avoid local optima and enhances optimization precision. Compared to random perturbations, this directional adjustment reduces unnecessary searches and accelerates convergence. The gradient descent-based adaptive guidance strategy smooths the optimization process through continuous small-scale adjustments, mitigating the disruptive effects of large random perturbations. The steps are as follows:**a** **Fitness Function**

For each point $P_{k,i}$ on the path of $p_{k}$, we define the fitness function $\mathrm{Fitness}(p_{k})$, which incorporates both curvature and smoothness penalties, as shown in Equation (Equation 20):(20)$$\mathrm{Fitness}\left(p_{k}\right)=\sum_{i=2}^{n-1}{\mathrm{curvature}}_{i}+{\mathrm{second}\_\mathrm{diff}}_{i}$$

This function embodies the trade-off between minimizing path curvature and optimizing smoothness, which is a crucial prerequisite for path optimization. It effectively reduces sharp turns and oscillations within the path.

**b** 
**Gradient Calculation**


To minimize the cost function $\mathrm{Fitness}(p_{k}\left[i\right])$, we need to compute its gradient. We can calculate the gradient of the curvature, ${\mathrm{curvature}}_{i}$, and the gradient penalty, ${\mathrm{second}\_\mathrm{diff}}_{i}$, for this purpose, as follows:*i* *Gradient of the Curvature*

First, we need to compute the gradients of $v_{i-1}$ and $v_{i}$, and then we use the chain rule to determine the gradient of the curvature, as shown in Equation (Equation 21):(21)$$\nabla {\mathrm{curvature}}_{i}=\frac{\partial {\mathrm{curvature}}_{i}}{\partial P_{k,i}}=\frac{\partial (1-\cos\left(\theta_{i}\right))}{\partial P_{k,i}}=-\frac{\partial \cos(\theta_{i})}{\partial P_{k,i}}$$
where the partial derivative of $\cos(\theta_{i})$ with respect to $P_{k,i}$ is given by Equation (Equation 22):(22)$$\frac{\partial \cos(\theta_{i})}{\partial P_{k,i}}=\frac{\partial \left(\frac{v_{i-1}\cdot v_{i}}{∥v_{i-1}∥∥v_{i}∥}\right)}{\partial P_{k,i}}$$

Due to the complexity of the calculation, we approximate the gradient numerically using the central difference formula in Equation (Equation 23):(23)$$\nabla {\mathrm{curvature}}_{i}\approx \frac{\mathrm{curvature}\left(P_{k,i}+\epsilon \right)-\mathrm{curvature}\left(P_{k,i}-\epsilon \right)}{2\epsilon }$$
where $\epsilon \in (10^{-6},10^{-4})$.

Numerical approximation maintains sufficient precision while considering computational efficiency, particularly excelling when step sizes are small.

*ii* 
*Gradient of the Smoothness Penalty*


The gradient of the smoothness penalty is given by Equation (Equation 24):(24)$$\nabla {\mathrm{second}\_\mathrm{diff}}_{i}=4P_{k,i}-2P_{k,i-1}-2P_{k,i+1}$$

**c** 
**Gradient Descent**


By iteratively updating each point on the path to minimize the fitness function $\mathrm{Fitness}(p_{k})$, the updated path $P_{k}{\left[i\right]}^{\prime }$ is represented by Equation (Equation 25):(25)$$P_{k}{\left[i\right]}^{\prime }=P_{k}\left[i\right]-\eta \left(g\right)\nabla \mathrm{Fitness}\left(p_{k}\left[i\right]\right)$$
where

1. η(g) is the exponentially decaying learning rate, defined by Equation (Equation 26):(26)$$\eta \left(g\right)=\eta_{0}\cdot e^{-\lambda \cdot g}$$
where:$\eta_{0}$: Initial learning rateλ: Decay rate*g*: Current iteration number

2. The gradient of the fitness function at each point is given by Equation (Equation 27):(27)$$\nabla \mathrm{Fitness}\left(p_{k}\left[i\right]\right)=\nabla {\mathrm{curvature}}_{i}+\nabla {\mathrm{second}\_\mathrm{diff}}_{i}$$

The importance of local adjustment in optimizing paths is reflected in the application of gradient descent strategies. By utilizing local gradient information, gradient descent incrementally improves the accuracy of the solution through local adjustments. An exponentially decaying learning rate allows the algorithm to make significant updates in the early stages and progressively refine the path as it approaches the optimal solution.

In this section, the variant of the COA integrates four major improvements over the traditional COA, including the initialization of the coati population using the honey badger algorithm, an enhanced heuristic function, a dynamic multi-population strategy, and a gradient descent-based adaptive guidance strategy. Subsequently, a novel variant of the COA is introduced, termed the novel adaptive coati optimization algorithm (NACOA). The pseudocode for the NACOA is presented in Algorithm 2, and its flowchart is illustrated in Figure 4.

The implementation of the algorithm involves the following five steps:Step 1: Algorithm Parameter Initialization: The algorithm begins by initializing a series of parameters, including $a_{1}$, β, $M_{0}$, $M_{min}$, $M_{max}$, and the number of iterations *G*. This initialization establishes the foundation for subsequent path optimization and sub-population operations.Step 2: HBA Path Optimization: During the path optimization phase, the algorithm uses the HBA path list as the initial input. For each iteration, the algorithm removes the first and last elements from the path list and deletes a specific point $p_{k}\left[i\right]$ using a randomly generated *t* value. The remaining points in the path are updated according to a specific equation (e.g., Equation (Equation 5)). After updating the path, the algorithm calculates the fitness function using a series of equations (e.g., Equations (6)–(11)) and determines the current optimal path.Step 3: NACOA Path List Initialization and Update: In the second stage, the algorithm initializes the NACOA path list based on the HBA path list, with $M_{g}=M_{0}$ and $N_{g}=\frac{N}{M_{0}}$. As the iterations progress, the number of sub-populations $M_{g}$ is dynamically adjusted according to Equation (16). During each update, the algorithm clears the current sub-population and assigns path points $pt_{i}$ to the sub-population $sub_{p}op_{a}$. The algorithm then optimizes the path using a gradient descent strategy and specified equations (Equations (12)–(14)), incorporating the optimized path points into the population list.Step 4: Sub-population Exchange and Global Optimization: At specific iteration counts (e.g., g%T==0), the algorithm randomly exchanges individuals between two sub-populations to increase diversity. At the end of each iteration, the algorithm identifies the current global best individual $g_{best}$ based on the fitness function.Step 5: Output Optimal Path: After all iterations are completed, the algorithm outputs the globally optimal path obtained through computation. This final output represents the optimal solution under the given constraints.
**Algorithm 2** The Pseudocode for the NACOA  1:Initialize various algorithm parameters, including $a_{1}$, β, $M_{0}$, $M_{\min}$, $M_{\max}$, *G*;  2:Initialize HBA path list;  3:**for** *k* from 1 to *n* do increasing *n* by 1 each time **do**  4:   path_current=path;  5:   Delete the first and last element of the path_current list;  6:   **for** *i* from 2 to n−1 do increasing *t* by 1 each time **do**  7:     t← randomly generate, and the range of *t* is between [1,n−i];  8:     $p_{k}\left[i\right]=path\_current\left[t\right]$, and $p_{k}$ is a member of populations;  9:     Delete path_current[t];10:     Update the points in the path according to Equation (Equation 5);11:   **end for**12:   Calculate the fitness function according to Equations (6)–(10);13:   Find the current optimal path according to Equation (Equation 11);14:**end for**15:Initialize NACOA path list = HBA path list, $M_{g}=M_{0}$, $N_{g}=\frac{N}{M_{0}}$, t=0, $\eta_{0}$;16:**for** *g* from 1 to *G* do increasing *G* by 1 each time **do**17:   The number of sub-populations $M_{g}$ was updated according to Equation (Equation 16);18:   population.clear();19:   **for** *a* from 1 to $M_{g}$ do increasing $M_{g}$ by 1 each time **do**20:     $sub\_pop_{a}.clear\left(\right)$;21:     **for** *k* from 1 to $N_{g}$ do increasing $N_{g}$ by 1 each time **do**22:        t=t+1;23:        $sub\_pop_{a}\left[k\right]=p_{t}$;24:        $p_{k}=sub\_pop_{a}\left[k\right]$;25:        Update the path according to Equations (12)–(14);26:        Gradient descent strategy is adopted to update the path according to Equation (Equation 25);27:     **end for**28:     Calculate the fitness function according to Equations (6)–(10);29:     Individual $p_{best_{a}}$ with the least fitness is found in $sub\_pop_{a}$ according to Equation (Equation 17);30:     $population.append(sub\_pop_{a})$;31:   **end for**32:   **if** g%T==0 **then**33:     Exchange two random individuals of two random sub_pop according to Equation (Equation 19);34:   **end if**35:   The global optimal individual $g_{best}$ is found according to Equation (Equation 18);36:**end for**37:Output the final optimal path.

## 5. Experiments

We evaluated our proposed hybrid network model and its four constituent base models using various performance metrics. For the unsmoothed paths identified by model features, we applied mean filtering, Bézier curve fitting, and the NACOA according to the level of roughness. This section presents the network experiments, results, and optimization experiments and results.

### 5.1. Network Experiments and Results

#### 5.1.1. Network Architecture

The structure of the hybrid network model MCRL, which combines the base neural network models MLP, CNN, RNN, and LSTM with a linear attention mechanism, is shown in Figure 5.

The model employs a decay function to initialize the optimizer with a learning rate, decaying by 10% every 1000 steps. The input vector size is 50×2, which is initially divided into four branches and processed through linear transformations.

The first branch utilizes the ReLU linear attention mechanism to process data, which are then fed into the MLP. Initially, the two-dimensional output vector is flattened using TensorFlow’s flatten module into a one-dimensional vector of size 100×1. This vector is then passed through four hidden layers with 128, 64, 32, and 16 nodes, each followed by a ReLU activation function, finally transforming the output vector to a size of 16×1. It then passes through a hidden layer with four nodes and a Softmax activation function to produce the output.

The second branch downsamples the data, converting the two-dimensional output vector to a size of 30×2. After processing with the ReLU linear attention mechanism, the data are fused with the MLP output and fed into the CNN unit. It first passes through a 3×3 convolutional layer with 32 kernels, followed by a 2×2 max-pooling layer, resulting in an output vector of size 15×32. This is followed by a dropout layer with a default probability of 0.2, then flattened into a 480×1 vector. The vector is then passed to a hidden layer with 512 nodes using ReLU activation and finally through a hidden layer with four nodes using Softmax activation to produce the output.

The third branch downsamples the data, converting the two-dimensional output vector to a size of 20×2. After processing with the ReLU linear attention mechanism, the data are fused with the CNN output and fed into the RNN unit. It passes through three hidden layers with 128, 64, and 64 nodes, with the third layer returning only the last timestep output, resulting in a 64×1 vector. Each layer is followed by a Tanh activation function. Finally, it passes through a hidden layer with four nodes using Softmax activation to produce the output.

The fourth branch downsamples the data, converting the two-dimensional output vector to a size of 10×2. After processing with the ReLU linear attention mechanism, the data are fused with the RNN output and fed into the LSTM unit. It passes through three hidden layers with 64, 32, and 32 nodes, with the third layer returning only the last timestep output, resulting in a 32×1 vector. Each layer is followed by a Tanh activation function. Finally, it passes through a hidden layer with four nodes using Softmax activation to produce the output.

The outputs of the four branches are finally fused and output through a linear transformation.

#### 5.1.2. Experimental Results

We utilized datasets comprising the machining G-code for aerospace gear, pentagram-shaped bosses, and maple leaf models. The MCRL model was employed to classify paths within these datasets. The experimental environment was an RTX 4070.

Hyperparameter settings

Table 2 presents the hyperparameter settings for all experiments. This table includes the initial learning rate, learning rate decay rate, decay steps, batch size, training iterations before convergence, and total training time (in minutes).

The hyperparameters were set as follows:Learning Rate: This parameter was set to either 0.0001 or 0.0005, depending on the dataset. The learning rates for the gear and pentagram datasets were 0.0001, while the rates for the maple leaf datasets were 0.0005.Decay Rate: The decay rate varied between datasets, ranging from 0.8 to 0.9. The decay rates for the gear datasets were higher at 0.9, indicating a slower reduction in the learning rate over time compared to the 0.8 rate for the pentagram and maple leaf datasets.Decay Steps: The decay steps were either 100 or 500. The gear and maple leaf datasets used a larger number of decay steps (500), implying a lower frequency of learning rate decay application, while the pentagram datasets employed 100 decay steps.Batch Size: The batch sizes were set to either 32 or 64. The gear and maple leaf datasets used a batch size of 64, which generally provides a more stable gradient estimate, whereas the pentagram dataset used a smaller batch size of 32.Iterations: The number of iterations required varied significantly between datasets. The gear dataset required the most iterations, totaling 497, while the pentagram dataset required 203. The maple leaf datasets required 234 and 340 iterations, respectively.Time: The training times for each dataset also varied. The gear dataset required the longest training time, at 212 units, while the pentagram dataset required 169 units, and the maple leaf dataset required the shortest time of 40 units.

These configurations highlight the tailored approach needed to optimize model performance across different datasets, demonstrating the variability of training parameters essential for achieving effective learning outcomes.

2.Performance Metrics

We evaluated the performance of the MCRL model using metrics such as accuracy, loss, precision, recall, F1 score, and AUC.

Accuracy: This metric represents the proportion of correctly predicted samples out of the total number of samples,
(28)$$\mathrm{Accuracy}=\frac{TP+TN}{TP+TN+FP+FN}$$
where:TP (True Positives): The number of samples that are truly positive and predicted as positive.TN (True Negatives): The number of samples that are truly negative and predicted as negative.FP (False Positives): The number of samples that are truly negative but predicted as positive.FN (False Negatives): The number of samples that are truly positive but predicted as negative.

Loss: This metric measures the disparity between the model’s predictions and the actual labels. In this experiment, the multiclass cross-entropy was used as the loss function.
(29)$$\mathrm{Categorical}\;\mathrm{Cross}-\mathrm{Entropy}\;\mathrm{Loss}=-\frac{1}{N}\sum_{i=1}^{N}\sum_{j=1}^{C}y_{i,j}\log\left(p_{i,j}\right)$$
where:*N* is the total number of samples.*C* is the total number of classes.$y_{i,j}$ is the true label of the *i*-th sample, where $y_{i,j}=1$ if the true label of the *i*-th sample is class *j*, and 0 otherwise.$p_{i,j}$ is the predicted probability that the *i*-th sample belongs to class *j*.$\log(p_{i,j})$ is the natural logarithm of the predicted probability of class *j*.

Precision: This metric indicates the proportion of correctly predicted positive samples out of all samples predicted as positive.
(30)$$\mathrm{Precision}=\frac{TP}{TP+FP}$$
High precision signifies that the model accurately predicts positive classes, with few false positives.

Recall: This metric measures the proportion of correctly predicted positive samples out of all actual positive samples.
(31)$$\mathrm{Recall}=\frac{TP}{TP+FN}$$
High recall indicates that the model can identify a majority of the actual positive samples.

F1 score: This metric measures the harmonic mean of precision and recall, providing a comprehensive measure of the model’s performance.
(32)$$F1\;\mathrm{Score}=\frac{2\times \mathrm{Precision}\times \mathrm{Recall}}{\mathrm{Precision}+\mathrm{Recall}}$$

AUC: The area under the ROC (Receiver Operating Characteristic) curve quantifies the model’s capability to distinguish between classes.
(33)$$\mathrm{AUC}=\int_{0}^{1}\mathrm{TPR}\left(\mathrm{FPR}\right)\;d\mathrm{FPR}$$
where:FPR: False Positive RateTPR: True Positive Rate

In this study, we compared the MCRL model with three other prominent models: ConvMixer (2022) [29], ConvNeXt (2022) [30], and MaxViT (2022) [31]. Table 3 presents the performance metrics of the various models across different datasets.

In assessing model performance across various datasets, our model (MCRL) exhibited outstanding results in multiple tests. A thorough analysis of the experimental outcomes from the gear, pentagram, and maple leaf datasets provided an in-depth understanding of the performance variations between MCRL and the other models across different datasets. These findings are crucial for evaluating the overall effectiveness of the models and their adaptability to various application contexts. The following section provides a detailed analysis of the performance of the different models on each dataset.


**Gear Dataset**


The results of the models on the gear dataset can be summarized as follows:The MCRL model achieved the best performance on the gear dataset, with an accuracy of 94.75%, a loss of 14.26, a precision of 96.23%, a recall of 93.52%, an F1 score of 94.85%, and an AUC of 97.71%. These results underscore MCRL’s superior predictive performance on this dataset and its ability to effectively balance various metrics.The ConvMixer and ConvNeXt models also performed relatively well, with accuracies of 92.22% and 93.92%, respectively. While their other metrics were comparable, they still fell short compared to MCRL.MaxViT slightly lagged behind ConvNeXt in overall performance but maintained high precision and F1 score.The MLP, CNN, RNN, and LSTM models showed mediocre performance, particularly the MLP and RNN models, which fell below 90% across all metrics, highlighting their limitations on this dataset.


**Pentagram Dataset**


The results of the models on the pentagram dataset can be summarized as follows:On the pentagram dataset, MCRL again excelled with an accuracy of 94.98%, a loss of 13.55, a precision of 96.47%, a recall of 93.35%, an F1 score of 95.63%, and an AUC of 97.58%.The ConvMixer and ConvNeXt models performed well on the pentagram dataset, although they slightly lagged behind MCRL in recall and F1 score, with values around 93% and 92%, respectively.MaxViT’s performance was fairly balanced, but it fell short in accuracy and AUC.Other models, such as MLP, CNN, and RNN, showed relatively poor performance, with RNN notably underperforming, achieving an accuracy of only 86.34% and significant deficiencies in recall and F1 score.


**Maple Leaf Dataset**


The results of the models on the maple leaf dataset can be summarized as follows:On the maple leaf dataset, the MCRL model achieved the highest accuracy of 96.32%, the lowest loss of 9.81, and precision and F1 scores of 96.52% and 96.35%, respectively, with an AUC of 98.66%. These results highlight MCRL’s exceptional performance in path classification tasks.The ConvMixer and ConvNeXt models followed closely, with accuracies of 95.97% and 95.27%, and F1 scores exceeding 95%, although they still fell slightly short of MCRL.MaxViT also showed stable performance, but its AUC was somewhat lower at 98.00.Traditional models, such as MLP, CNN, and RNN, generally fell short of the advanced models mentioned above, with RNN showing particularly poor performance on this dataset, achieving an accuracy of only 85.63%.

Overall, the MCRL model exhibited exceptional performance across all four datasets, particularly demonstrating the best comprehensive metrics on the maple leaf dataset. Other models, such as ConvMixer, ConvNeXt, and MaxViT, also performed admirably, making them well suited for tasks requiring high accuracy and precision. In contrast, traditional models like MLP, CNN, RNN, and LSTM performed relatively poorly on these datasets, with RNN, in particular, struggling to match the performance of more advanced models in complex path classification tasks.

Accuracy and loss are the two most crucial evaluation metrics, as they provide a clear reflection of the model’s performance and optimization status. Figure 6 and Figure 7 illustrate the accuracy and loss of various models across different datasets.

Figure 6 compares the performance of various models, including MCRL, ConvMixer, ConvNeXt, MaxViT, MLP, CNN, RNN, and LSTM, across each dataset. The x-axis represents the number of training epochs, ranging from 0 to 500, while the y-axis displays the accuracy, which ranges approximately from 0.72 to 0.96.

Across all datasets, the MCRL model (depicted by the red line) consistently demonstrated superior performance, achieving the highest accuracy early in training and maintaining this advantage throughout subsequent epochs. ConvMixer, ConvNeXt, and MaxViT also performed admirably, although their accuracies were generally slightly lower than that of MCRL. In contrast, traditional models, such as MLP, CNN, RNN, and LSTM (represented by lighter colors), typically exhibited lower accuracy and slower convergence.

This comparative chart highlights MCRL’s exceptional performance across various datasets, illustrating its robustness and efficacy in different scenarios. The convergence patterns indicate that while most models stabilized after around 100 training epochs, MCRL excelled in both convergence speed and final accuracy, particularly on the gear and maple leaf datasets.

Figure 7a presents the loss curves of different models on the gear dataset as the training epochs progressed. It is evident that the MCRL model (red curve) converged more rapidly than the other models, achieving the lowest final loss value. Other models, such as ConvMixer, ConvNeXt, MaxViT, MLP, CNN, RNN, and LSTM, also exhibited a downward trend in loss during early training, but their final loss values were higher than those of MCRL.

In Figure 7b, it is evident that the loss values for all models decreased as the number of training epochs increased. The MCRL model once again demonstrated the fastest convergence rate and the lowest final loss value, with other models trailing behind. Notably, MaxViT and LSTM exhibited a slower rate of loss reduction and ended with higher loss values compared to the other models.

In Figure 7c, it is evident that the MCRL model continued to exhibit the best convergence speed and the lowest final loss value. The performance of other models mirrored the trends observed in the previous datasets: a rapid initial decrease in loss followed by stabilization, but with final loss values higher than those of MCRL.

Overall, the MCRL model consistently demonstrated superior convergence speed and lower final loss values across all datasets, outperforming the other models.

3.Inference Time

Inference time refers to the duration a model requires to process input data and produce output predictions. It is a critical metric that directly impacts resource efficiency in deployment scenarios. Quantization-aware training (QAT) achieves this by simulating the effects of quantization during the training process, reducing the network’s weights and activations from 32-bit floating-point numbers to lower-precision 8-bit integers. This reduction in precision results in smaller memory usage, which accelerates data transfer and reduces memory bandwidth demands. Consequently, this experiment employed the QAT strategy to diminish the model’s inference time. Table 4 presents the inference times (in seconds) of the proposed model alongside those of ConvMixer, ConvNeXt, MaxViT, MLP, CNN, RNN, and LSTM.

The observations of the inference time results are summarized as follows:MCRL: The inference times across various datasets ranged from 3.1 to 3.9 s.MCRL with QAT: This variant of MCRL utilized quantization-aware training (QAT) technology, which simulates the effects of reduced precision during training, lowering network weights and activations from 32-bit floating-point numbers to a minimum of 8-bit integers. This reduction in precision decreased memory usage and accelerated data transfer, thereby reducing inference time compared to the standard MCRL model. For instance, on the gear dataset, the inference time for MCRL with QAT was 2.8 s, while the standard MCRL took 3.2 s. Similar improvements were observed across all datasets.ConvMixer (2022), ConvNeXt (2022), MaxViT (2022): These models, introduced in 2022, achieved inference times comparable to MCRL but showed variability across datasets, with ConvMixer and RNN exhibiting slightly higher inference times.MLP, CNN, RNN, LSTM: These traditional models generally had higher inference times, with MLP showing the highest inference times across all datasets, particularly on the pentagram and maple leaf datasets.

In summary, MCRL with QAT emerged as a strong contender among the evaluated models, optimizing inference time while maintaining performance.

Taking the paths in the gear dataset as an example, four curves were randomly selected based on categories, and their classification accuracy within the network was examined. The green lines represent correct classifications, while the red lines indicate misclassifications, as shown in Table 5.

The analysis revealed that the MCRL model achieved the best overall classification performance, with an impressive record of complete accuracy. In contrast, the ConvMixer, ConvNeXt, MaxViT, MLP, CNN, RNN, and LSTM models each exhibited a misclassification in one of the selected curves.

In summary, MCRL combines the strengths of MLP, CNN, RNN, and LSTM, significantly enhancing the overall performance of the model and making it the most effective deep learning network model for the given task.

### 5.2. Optimization Experiments and Results

#### 5.2.1. Complexity Analysis

The computational complexity of algorithms can be divided into two principal components. The first component arises from path updates and node operations, with a total complexity of $O(n^{2})$, where *n* denotes the number of path points. This signifies that during the processing of all path points, the complexity escalates to a quadratic level due to the presence of nested loops. The second component relates to the iterative process of the algorithm, encompassing population updates and the execution of gradient descent, with a complexity of O(t×p×d), where *t* represents the maximum number of iterations, *p* denotes the population size, and *d* signifies the dimensionality of the problem. Overall, the algorithm’s total computational complexity is $O(n^{2})$ + O(t×p×d).

#### 5.2.2. Comparison with Other Algorithms

In this study, we compared the NACOA with three other major algorithms (Table 6): the African Vultures Optimization Algorithm (AVOA) [32], Sand Cat Swarm Optimization (SCSO) [33], and the Egret Swarm Optimization Algorithm (ESOA) [34].

The runtime of the NACOA was 10 min, demonstrating superior performance in the time dimension compared to other algorithms, second only to the ESOA (11 min). This indicates that the NACOA can identify optimal solutions within a relatively short timeframe. The GPU utilization of the NACOA was 16%, the lowest among the four algorithms, showcasing its efficiency in terms of hardware resource consumption. In contrast, the AVOA exhibited a GPU utilization rate as high as 23%. The complexity of the NACOA, represented as $O\left(n^{2}\right)+O\left(t\times p\times d\right)$, highlights its superiority in addressing large-scale problems. This complexity ensures the stable performance of the NACOA in practical applications. We consider the following formula, defining efficiency as
(34)$$\mathrm{Efficiency}=\frac{\mathrm{Theoretical}\;\mathrm{Optimal}\;\mathrm{Resource}\;\mathrm{Usage}-\mathrm{Actual}\;\mathrm{Resource}\;\mathrm{Usage}}{\mathrm{Time}}$$

In this formula, we account for the reduction in resource usage (theoretical value minus actual value) in relation to time. This formula indicates that if a greater reduction in resource usage is achieved within a shorter period, the efficiency will be higher. According to this formula, the NACOA achieved the highest efficiency, reaching a value of 8.4, while the efficiencies of the other algorithms fell below that of the NACOA.

#### 5.2.3. Comparison of Algorithm Optimization Details

Figure 8 illustrates the differences in tool-path optimization across various workpieces using different algorithms. In all three cases, the COA significantly outperformed the other algorithms, producing optimized paths that were both smoother and more closely aligned with the original tool path. This attribute is crucial in precision machining and other manufacturing processes where maintaining design fidelity is paramount. The smoothness of the COA path reduces mechanical stress and mitigates the risk of tool wear, potentially extending tool life and enhancing overall process efficiency.

In summary, the COA excels in providing smoother trajectories while preserving the integrity of the original path, making it an efficient tool for optimizing paths in complex and precision-critical applications.

#### 5.2.4. The Performance of Algorithms in Path Optimization

Figure 9, Figure 10 and Figure 11 illustrate the application of mean filtering, Bézier curve fitting, and the NACOA on randomly selected paths with varying degrees of irregularity. A comparison of the curves before and after optimization using mean filtering shows a 54% reduction in curvature. Similarly, the Bézier curve fitting resulted in a 67% decrease in curvature, while the NACOA achieved a more substantial improvement, reducing curvature by 78%.

Overall, the optimized curves significantly improve the smoothness of the machining model, effectively reducing machine tool vibrations.

### 5.3. Integration of the Preprocessing Module in the CNC System

In this study, we propose the introduction of a preprocessing module into the CNC system, designed to analyze the key information embedded in G-code through intelligent methods. Collaborating with existing CAM software, this module performs preprocessing before the execution of G-code. Following the initial analysis, the system employs a line attention mechanism and the deep learning model MCRL to identify and categorize potential tool-path regions where machining errors may arise due to discontinuous curvature. The system then automatically selects and applies the appropriate optimization techniques based on the detected irregularities.

For the end user, the integration process is nearly seamless, requiring minimal manual intervention while ensuring greater machining accuracy and smoother tool paths. The reduction in machining errors will enhance output quality, minimize rework, and improve overall efficiency. These advancements in the CNC workflow promise significant time and cost savings, making the machining process more reliable and user-friendly.

## 6. Conclusions

This paper introduces a deep learning method for feature recognition of unsmooth paths, employing filtering, fitting, and the proposed NACOA to address sharp corner issues based on the degree of roughness. The MCRL deep learning network model integrates MLP, CNN, RNN, and LSTM models along with a linear attention mechanism. We evaluated the model’s performance using various metrics to demonstrate its substantial advancements. Experimental validation on G-code datasets for gear, pentagram, and maple leaf machining confirmed that the proposed model is highly effective in accurately identifying unsmooth paths, achieving a classification accuracy of 95.56%. In four randomly selected path curves, MCRL achieved perfect classification, while the other models exhibited varying degrees of misclassification. Further research indicated that MCRL surpassed its constituent models in terms of precision.

We anticipate that the developed model and algorithm can be implemented in machining environments to more accurately identify and optimize unsmooth paths. Depending on the identified degree of path roughness, we utilized mean filtering, Bézier curve fitting, and the NACOA for sharp corner recognition. Visual comparisons showed that the optimized paths were smoother than the original paths, effectively reducing machine tool vibrations.

In the future, we plan to adopt more advanced deep learning network integration techniques and optimization algorithms, incorporating a broader range of workpiece machining G-code datasets to further refine our proposed MCRL model and NACOA. Additionally, we aim to apply interpretable AI techniques to gain deeper insights into the decision-making processes of the MCRL model and NACOA, thereby enhancing the accuracy of unsmooth path recognition and the smoothness of algorithm optimization.

## Figures and Tables

**Figure 1 sensors-24-05720-f001:**
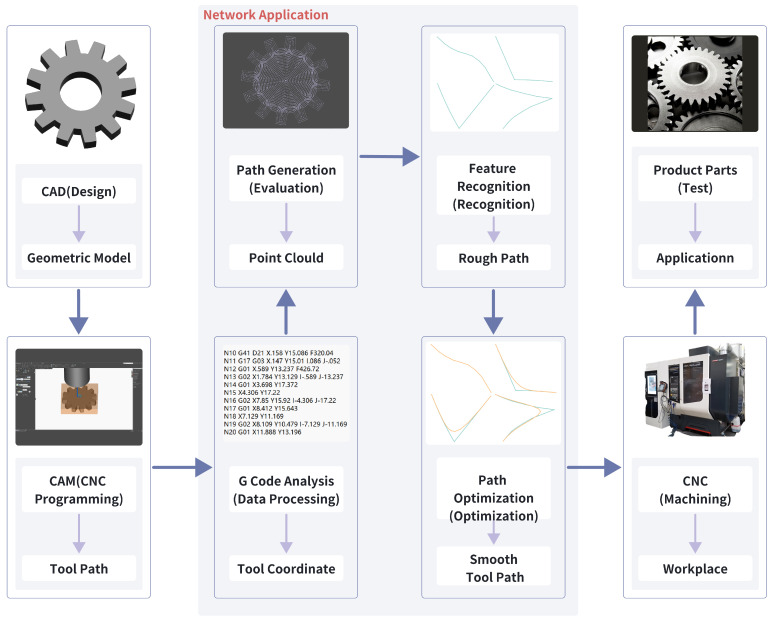
Web application for communication between CAM and machine tools.

**Figure 2 sensors-24-05720-f002:**
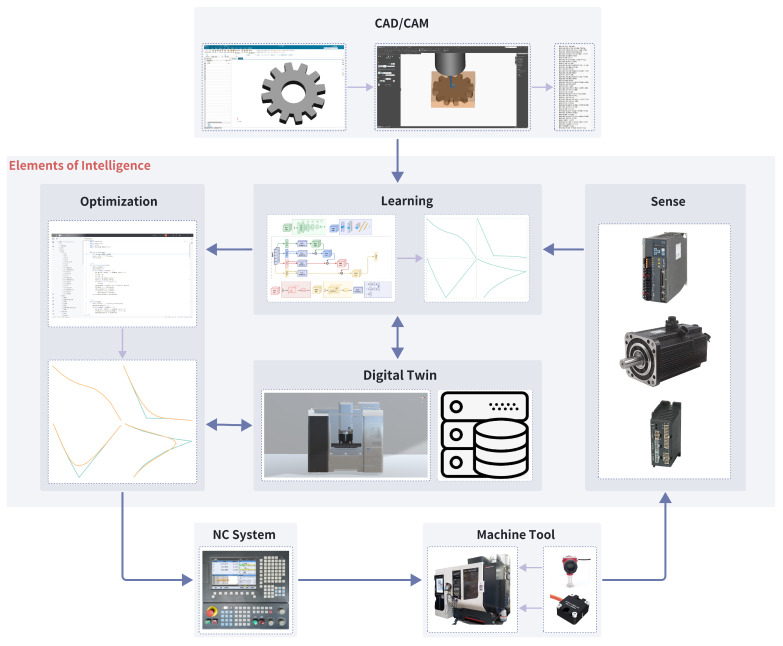
Modeling scheme of intelligent elements in a CNC system.

**Figure 3 sensors-24-05720-f003:**
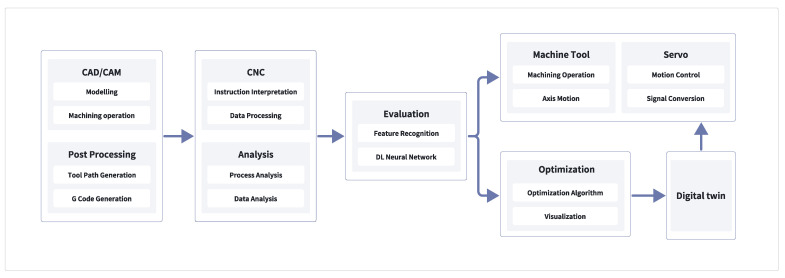
Machining process flow structure of an intelligent CNC system.

**Figure 4 sensors-24-05720-f004:**
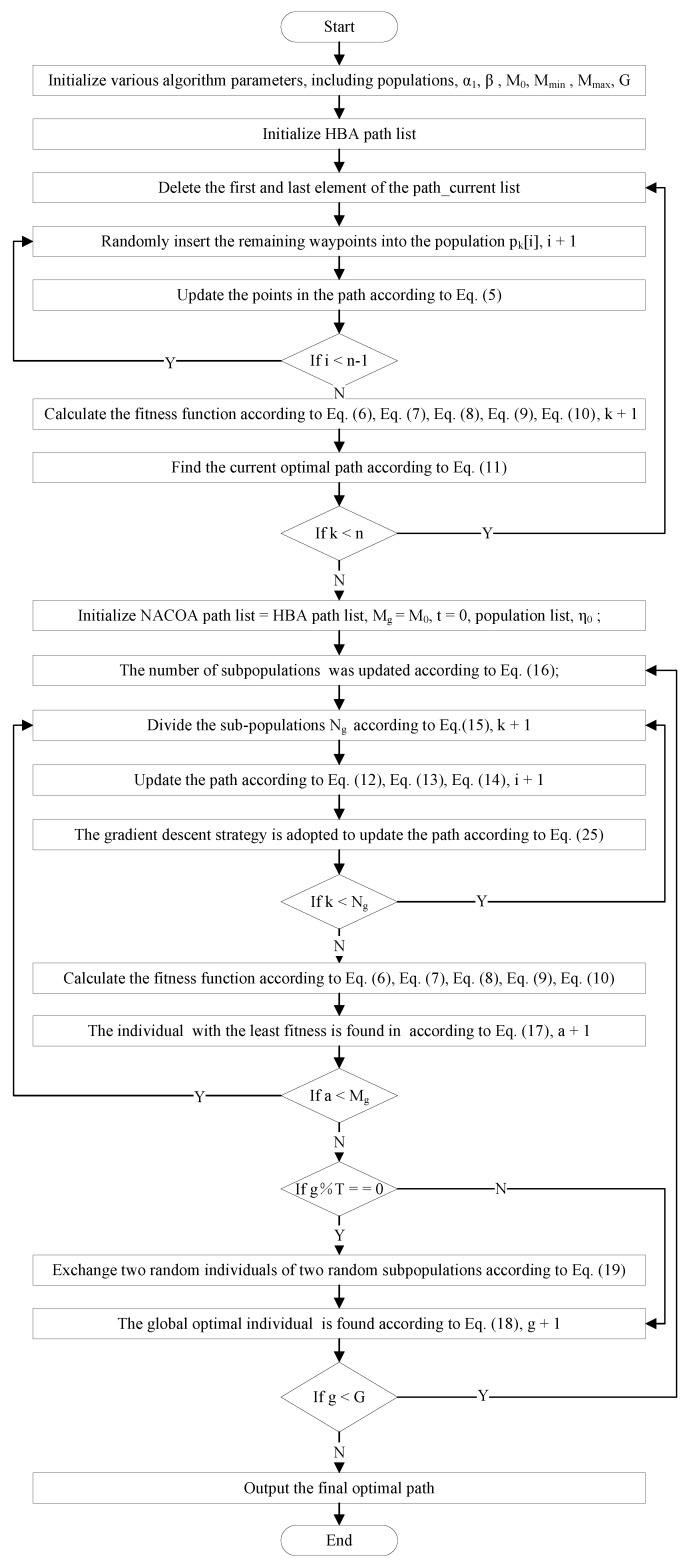
The flow chart of the NACOA.

**Figure 5 sensors-24-05720-f005:**
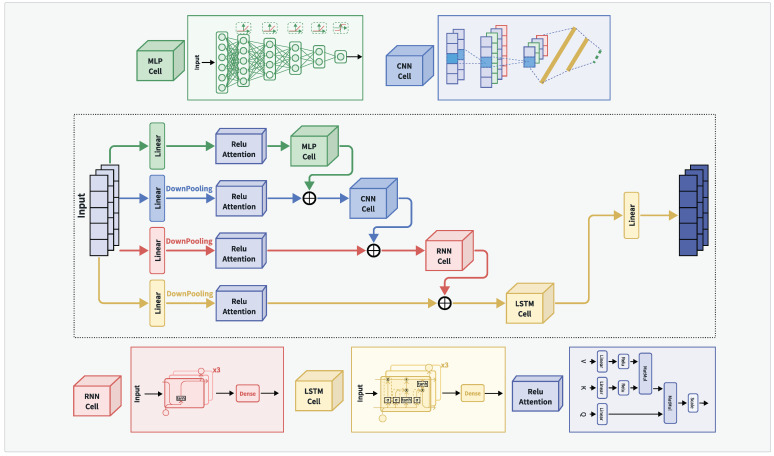
MCRL network model structure.

**Figure 6 sensors-24-05720-f006:**
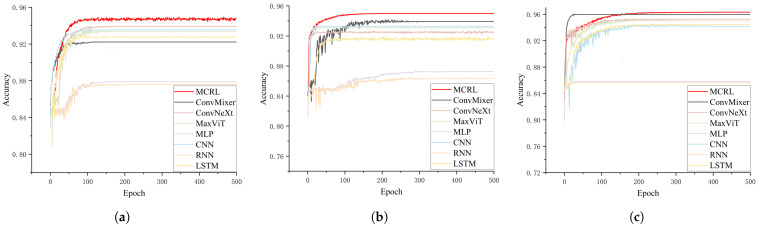
Accuracies of different models across different datasets. (**a**) Accuracies of models on the gear dataset. (**b**) Accuracies of models on the pentagram dataset. (**c**) Accuracies of models on the maple leaf dataset.

**Figure 7 sensors-24-05720-f007:**
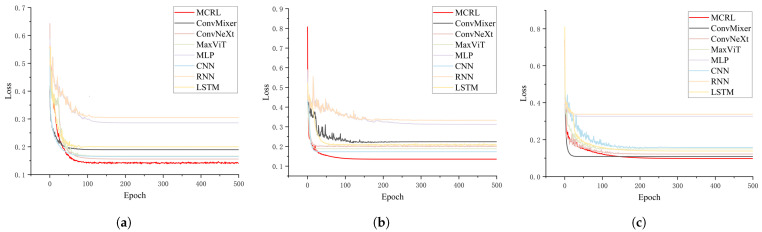
Losses of different models on different datasets. (**a**) Losses of models on the gear dataset. (**b**) Losses of models on the pentagram dataset. (**c**) Losses of models on the maple leaf dataset.

**Figure 8 sensors-24-05720-f008:**

Comparative analysis of path optimization algorithms. (**a**) Comparative analysis of path optimization algorithms on gear tool path. (**b**) Comparative analysis of path optimization algorithms on pentagram tool path. (**c**) Comparative analysis of path optimization algorithms on maple leaf tool path.

**Figure 9 sensors-24-05720-f009:**
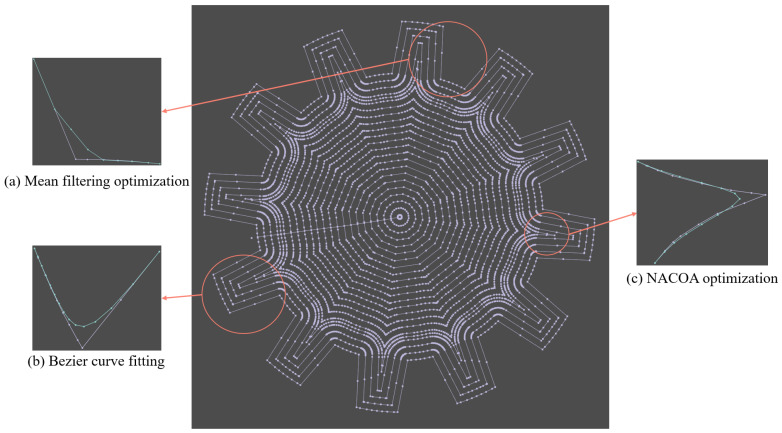
Path optimization results on the gear model.

**Figure 10 sensors-24-05720-f010:**
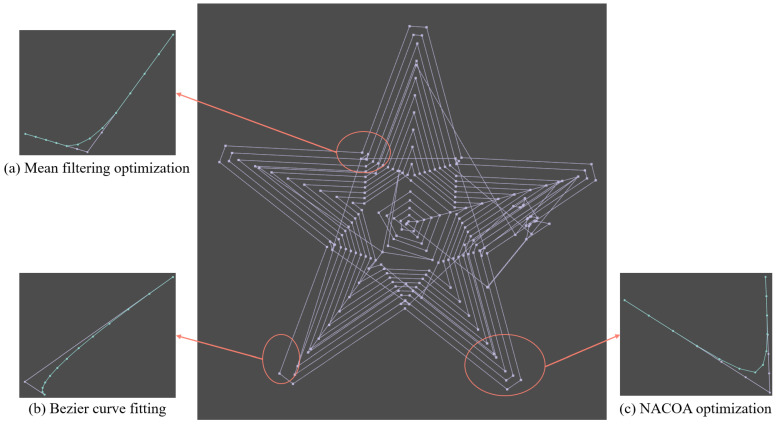
Path optimization results on the pentagram model.

**Figure 11 sensors-24-05720-f011:**
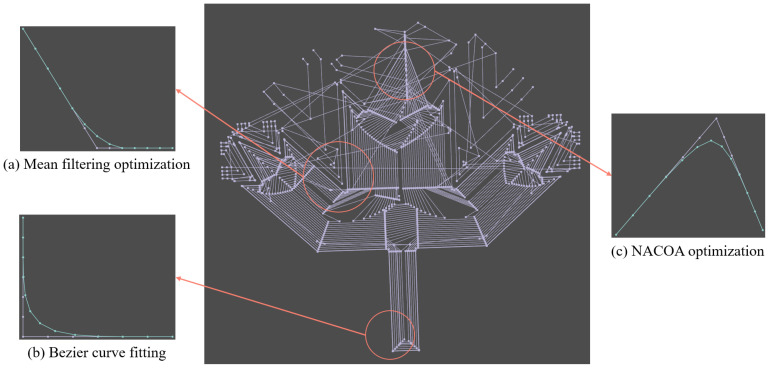
Path optimization results on the maple leaf model.

**Table 1 sensors-24-05720-t001:** Path roughness categories.

LocalCurvature	Description	Category	Evaluation
LocalCurvature ≤ 1.9	Smooth	Category 1	Good
1.9 < LocalCurvature < 3.5	Slightly Rough	Category 2	Moderate
3.5 < LocalCurvature < 7.6	Rugged	Category 3	Poor
LocalCurvature ≥ 7.6	Sharp Turning Corner	Category 4	Very Poor

**Table 2 sensors-24-05720-t002:** Hyperparameter settings for different datasets.

Dataset	Learning Rate	Decay Rate	Decay Steps	Batch Size	Epochs	Time
Gear	0.0001	0.9	500	64	497	212
Pentagram	0.0001	0.8	100	32	203	169
Maple Leaf	0.0005	0.8	500	64	340	140

**Table 3 sensors-24-05720-t003:** Performance metrics of different models across datasets.

Dataset	Model	Accuracy	Loss	Precision	Recall	F1 Score	AUC
Gear	MCRL	94.75	14.26	96.23	93.52	94.85	97.71
	ConvMixer (2022)	92.22	18.94	93.87	91.06	92.44	96.77
	ConvNeXt (2022)	93.92	15.60	95.54	91.92	93.69	98.15
	MaxViT (2022)	93.35	16.64	95.23	91.16	93.15	97.86
	MLP	87.90	28.61	94.50	82.35	88.00	93.55
	CNN	93.54	16.49	94.79	92.16	93.45	97.65
	RNN	87.59	30.54	93.86	88.86	91.29	92.77
	LSTM	92.77	20.00	94.09	91.66	92.85	96.97
Pentagram	MCRL	94.98	13.55	96.47	94.81	95.63	97.94
	ConvMixer (2022)	93.91	22.39	94.71	93.35	94.02	98.04
	ConvNeXt (2022)	92.50	20.07	95.25	92.77	93.99	96.47
	MaxViT (2022)	93.08	18.85	96.32	93.79	95.03	96.43
	MLP	87.26	31.23	92.26	87.13	89.62	92.25
	CNN	93.29	17.32	95.02	92.29	93.63	97.13
	RNN	86.34	33.30	91.27	81.36	86.03	90.42
	LSTM	91.67	20.84	93.96	92.72	93.33	96.40
Maple Leaf	MCRL	96.32	9.81	96.52	96.19	96.35	98.85
	ConvMixer (2022)	95.97	10.87	96.45	95.47	95.95	98.68
	ConvNeXt (2022)	95.27	12.22	95.34	95.08	95.20	98.48
	MaxViT (2022)	95.00	15.02	95.41	94.56	94.98	98.00
	MLP	85.81	32.66	90.73	80.52	85.32	90.69
	CNN	94.17	15.78	96.42	95.43	95.92	97.83
	RNN	85.63	33.78	90.71	80.25	85.16	90.01
	LSTM	94.47	13.85	95.03	93.86	94.44	98.31

**Table 4 sensors-24-05720-t004:** Inference times of different models on different datasets.

Model	Datasets		
	Gear	Pentagram	Maple Leaf
MCRL	3.2	3.1	3.9
MCRL with QAT	2.8	2.9	3.5
ConvMixer (2022)	3.7	3.8	3.7
ConvNeXt (2022)	3.7	3.2	3.8
MaxViT (2022)	3.0	3.2	4.1
MLP	4.1	4.6	4.6
CNN	3.7	3.5	4.3
RNN	3.7	3.8	4.5
LSTM	3.3	3.5	4.2

**Table 5 sensors-24-05720-t005:** Classification results of random curves by different network models.

	Category 1	Category 2	Category 3	Category 4
MCRL	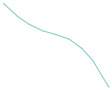	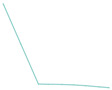	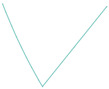	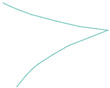
ConvMixer	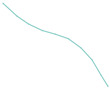	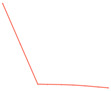	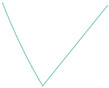	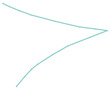
ConvNeXt	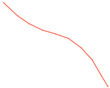	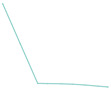	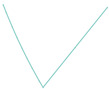	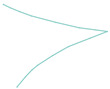
MaxVit	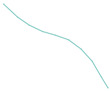	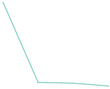	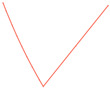	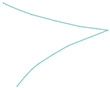
MLP	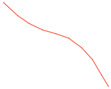	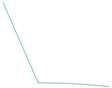	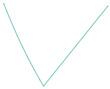	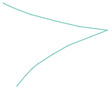
CNN	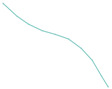	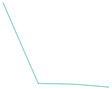	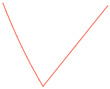	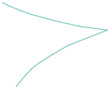
RNN	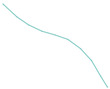	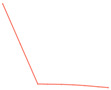	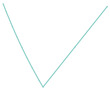	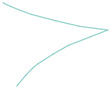
LSTM	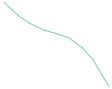	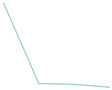	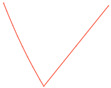	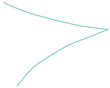

**Table 6 sensors-24-05720-t006:** Comparison with other algorithms.

	NACOA	SCSO	ESOA	AVOA
Time	10 min	12 min	11 min	17 min
GPU usage	16%	20%	18%	23%
Complexity	$O\left(n^{2}\right)+O\left(t\times p\times d\right)$	O(t×p×d)	O(t×p×d)	O(t×p)+O(t×p×d)
Efficiency	8.4	6.67	7.45	4.53
GPU	RTX 4070	RTX 4070	RTX 4070	RTX 4070

## Data Availability

The data presented in this study are available upon request from the corresponding author. They are restricted to the experimental results.

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
