# Peer review of "An Agent-Based Method for Feature Recognition and Path Optimization of Computer Numerical Control Machining Trajectories"

_sensors, 2024, doi:10.3390/s24175720_

Round 1

Reviewer 1 Report

Comments and Suggestions for Authors

- The writing and presentation should be improved. First, the authors should elaborate more on the underlying intuition or motivation behind the  CNC Machining Trajectories.

- The references in this article do not conform to the standard format, with some formally published works being cited as on-line versions.

- The compared methods are very old. The authors will need to consider recent methods.

- Please add and discuss some of the mentioned references. What is the advantage/drawback of the presented approach?

- Fine-tuning the parameters should be discussed.

- The paper lacks clarity in describing the specific methodologies and technical details employed in the proposed agent-based method.

- The evaluation metrics used in comparing the proposed model with existing ones are not thoroughly discussed, which raises concerns about the validity and comprehensiveness of the performance evaluations.

- The paper mentions the low reasoning speed of the proposed model without providing insights into potential optimization strategies or comparative analysis with other models.

- The language and organization of the paper could be improved for better readability and comprehension, particularly in explaining the rationale behind certain design choices and experimental setups.

Comments on the Quality of English Language

- The language and organization of the paper could be improved for better readability and comprehension, particularly in explaining the rationale behind certain design choices and experimental setups.

Reviewer 2 Report

Comments and Suggestions for Authors

This paper explores the application of artificial intelligence technology in the field of intelligent manufacturing, particularly in the optimization of CNC machining trajectories. Overall, this research work is meaningful and innovative, but needs to be modified:

1.      The effectiveness of the proposed algorithm was verified in the paper, but there is a lack of in-depth analysis on the efficiency of the algorithm, such as computation time and resource consumption. Suggest supplementing algorithm complexity analysis and efficiency comparison experiments.

2.      Although the experimental part validated the effectiveness of the method, further comparative experiments can be added, such as comparing it with other existing methods, to demonstrate the superiority of the proposed method.

3.      The experiments in the paper are mainly based on the G-code dataset of gear models. It is suggested to explore the applicability of this method in different types of CNC machining and a wider range of datasets.

4.      The paper can discuss how the proposed method can be integrated into existing CNC system workflows and its potential impact on end users.

5.      Some figures may require higher resolution to ensure clarity.

6.      In the symbol explanation section, the first letter of "where" should be lowercase, please note.

7.      The overall writing quality of the paper is high, but there are still some minor grammar and spelling errors. It is recommended to conduct a thorough proofreading round.

8.      In "Related Work " section, I feel the current coverage of the state of the intelligent manufacturing. is not satisfactory as the related work section does not cover many contributions that likely provide the building blocks of the proposed approach.

For example,

(1) LCDL: Towards Dynamic Localization for Autonomous Landing of Unmanned Aerial Vehicle Based on LiDAR-Camera Fusionï¼›

(2) Value-Driven Robotic Digital Twins in Cyber–Physical Applicationsï¼›

(3) LiDAR Depth Cluster Active Detection and Localization for a UAV with Partial Information Loss in GNSSï¼›

It is suggested to cite the above articles and analyze the differences in Section Related Works.

Comments on the Quality of English Language

The overall writing quality of the paper is high, but there are still some minor grammar and spelling errors. It is recommended to conduct a thorough proofreading round.

Round 2

Reviewer 1 Report

Comments and Suggestions for Authors

In the revised manuscript the authors carefully addressed the raised questions and concerns. The manuscript contains now all information and can be accepted for publication. I believe that the changes the authors made enhanced an already fine paper, and I am pleased to inform that this manuscript for publication in the journal.

Comments on the Quality of English Language

In the revised manuscript the authors carefully addressed the raised questions and concerns. The manuscript contains now all information and can be accepted for publication. I believe that the changes the authors made enhanced an already fine paper, and I am pleased to inform that this manuscript for publication in the journal.

Reviewer 2 Report

Comments and Suggestions for Authors

The author has completed the revisions based on the feedback and is now ready for publication in its current form.